# Differentiable Model Predictive Control on the GPU

**Emre Adabag, Marcus Greiff, John Subosits, Thomas Lew**
Toyota Research Institute

## Abstract

Differentiable model predictive control (MPC) offers a powerful framework for combining learning and control. However, its adoption has been limited by the inherently sequential nature of traditional optimization algorithms, which are challenging to parallelize on modern computing hardware like GPUs. In this work, we tackle this bottleneck by introducing a GPU-accelerated differentiable optimization tool for MPC. This solver leverages sequential quadratic programming and a custom preconditioned conjugate gradient (PCG) routine with tridiagonal preconditioning to exploit the problem's structure and enable efficient parallelization. We demonstrate substantial speedups over CPU- and GPU-based baselines, significantly improving upon state-of-the-art training times on benchmark reinforcement learning and imitation learning tasks. Finally, we showcase the method on the challenging task of reinforcement learning for driving at the limits of handling, where it enables robust drifting of a Toyota Supra through water puddles.

Code: https://github.com/ToyotaResearchInstitute/diffmpc
Video: https://youtu.be/r42iJBw-L4E

## 1 Introduction

Differentiable optimization tools enable leveraging the structured and precise outputs of optimization algorithms as inductive biases in machine learning, reducing data requirements and enforcing constraints. These methods also enable data-driven tuning of optimization algorithms, reducing time-consuming manual expert-driven development using data. In particular, differentiable model predictive control has many applications, such as in motion planning (Bhardwaj et al., 2020), parameter estimation (Landry et al., 2019) and tuning (Cummins et al., 2025), reinforcement learning (Romero et al., 2024), imitation learning (Grandia et al., 2023; Wan et al., 2024), and end-to-end planning and control (Amos et al., 2018; Karkus et al., 2023).

However, the adoption of differentiable optimization is hindered by the sequential nature of optimization tools: Efficient parallelization on graphics processing units (GPUs) remains difficult, both for solving the optimization problem (the forward pass) and for computing gradients (the backward pass). Recent methods for differentiable optimal control, such as (Bambade et al., 2024; Frey et al., 2025) only run on central processing units (CPUs) to leverage the time-induced sparsity of optimal control problems (OCPs) via sequential algorithms such as the iterative linear quadratic regulator (iLQR) (Amos et al., 2018). Overcoming this computational bottleneck could enable scaling up to large datasets and expressive architectures, fully utilizing the benefits of modern deep learning.

**Contributions**: We propose **Diff**erentiable **M**odel **P**redictive **C**ontrol (`DiffMPC`): a new tool for *differentiable optimal control* to solve and differentiate through optimal control problems of the form

$$\textbf{OCP}: \quad \underset{z=(x,u)}{\arg\min} \ \sum_{t=0}^{T} c_t^{x,\theta}(x_t) + \sum_{t=0}^{T-1} c_t^{u,\theta}(u_t) \ \text{ s.t. } \ f_t^\theta(x_{t+1}, x_t, u_t) = 0, \quad x_0 = x_s^\theta,$$

where the costs, equality constraints, and initial conditions are parametrized by parameters $\theta$, which could represent the weights of a neural network (parameterizing the cost $c(\cdot)$ or constraints $f(\cdot)$), the outputs of an intermediate neural network layer, or the parameters of a physics-based model. `DiffMPC` is tailored for execution on the GPU by leveraging the structure of **OCP**: The core of the solver is a preconditioned conjugate gradient (PCG) routine introduced in (Adabag et al., 2024) to

solve the linear system arising from the optimality conditions of **OCP**. This routine leverages the sparse structure of **OCP** to expose parallelism over time $t$, and enables warm-starting across problem instances. `DiffMPC` is written in `JAX` to simplify deployment in machine learning applications. Numerical and experimental results using `DiffMPC` show the following:

- **Significant acceleration on the GPU**: Comprehensive benchmarking against the state-of-the-art differentiable optimization libraries `mpc.pytorch` (Amos et al., 2018), `trajax` (Frostig et al., 2021), and `Theseus` (Pineda et al., 2022) show consistent speedups of more than $4\times$ when solving and differentiating **OCP**. These experiments include reinforcement learning (RL) and imitation learning (IL) examples, which are common applications of differentiable optimal control. These speedups primarily come from a PCG routine tailored for GPU execution and several related design choices in `DiffMPC` that exploit problem structure for parallelization.

- **Reliably drifting a Toyota Supra via domain randomization and reinforcement learning**: We use `DiffMPC` to automatically tune an MPC controller for driving at the limits of handling and ensure robustness to model mismatch, such as water puddles on the road. Specifically, we use domain randomization over the nonlinear dynamics to learn cost and vehicle parameters for the controller via reinforcement learning. In this application, due to the unstable nature of driving at the limits of handling where tire forces are saturated and the vehicle is prone to spinning out, large batches are conducive to robust training, motivating the use of the proposed GPU-accelerated differentiable optimization framework. Results demonstrate significantly improved performance on a Toyota Supra drifting through water puddles (Figure 7).

## 2 BACKGROUND ON DIFFERENTIABLE OPTIMIZATION

### 2.1 RELATED WORK

Recent years have seen a surge of works on scalable optimization algorithms that run efficiently on the GPU (Schubiger et al., 2020; Chen et al., 2024; Kang et al., 2024; Feng et al., 2024; Bishop et al., 2024; Adabag et al., 2024; Chari et al., 2024; Jeon et al., 2024; Amatucci et al., 2025; Montoison & Caillau, 2025), enabling large-scale validation and parameter sweeps. However, as optimization algorithms are inherently sequential, GPU-accelerated solvers are often slower than CPU-based solvers when solving small- to medium-sized problems individually. Among these efforts, `MPCGPU` (Adabag et al., 2024) exploits the sparse-in-time structure of **OCP** using a tailored preconditioned conjugate gradient (PCG) routine. As a result, it achieves fast replanning times for MPC applications that match or outperform state-of-the-art CPU-based solvers, even for single problem instances. This work highlights the importance of taking advantage of problem structure to fully exploit GPU parallelism. Still, these solvers are not differentiable and thus do not provide sensitivities with respect to problem parameters.

Differentiable optimization tools address this gap by enabling the computation of gradients of solutions to optimization problems with respect to parameters, supporting a wide range of applications (see Section 1). These tools fall into two main categories, general-purpose solvers (Ren et al., 2023; Arnold et al., 2020; Deleu et al., 2019; Pineda et al., 2022; DeepMind et al., 2020; Blondel et al., 2022) and structure-exploiting solvers for MPC (Amos et al., 2018; Agrawal et al., 2019; Frostig et al., 2021; Howell et al., 2022; Bambade et al., 2024; Frey et al., 2025).

General-purpose differentiable solvers often support GPU execution and can be applied broadly. However, in MPC applications, they are typically slower than **OCP**-specific solvers, as they do not sufficiently leverage problem structure. For example, solving **OCP** using `Theseus` (Pineda et al., 2022) by rolling out control inputs through the dynamics as in (Wan et al., 2024) can be orders of magnitude slower than using `DiffMPC` (see Section 4).

On the other hand, **OCP**-tailored solvers exploit problem structure for efficiency, but usually only support CPU execution (Amos & Kolter, 2017; Agrawal et al., 2019; Howell et al., 2022; Bambade et al., 2024; Frey et al., 2025), which can limit their scalability in large-batch or learning applications. The closest methods to `DiffMPC` are `mpc.pytorch` (Amos et al., 2018) and `trajax` (Frostig et al., 2021), which support GPU execution and exploit the time-induced sparse structure of **OCP**. However, both methods rely on Riccati recursions over time $t = 0, \dots, T$, requiring relatively large batch sizes to realize speedups on the GPU. As shown in Section 4, `DiffMPC` avoids these recursions using a tailored PCG routine, enabling significant speedups over existing methods.

## 2.2 A PRIMER ON DIFFERENTIABLE OPTIMIZATION

Next, we provide background on differentiable optimization (DO) used in the design of `DiffMPC`. Consider the generic parametric, equality-constrained, optimization problem

$$\mathbf{P}: \quad z = \arg\min_{z \in \mathbb{R}^n} f(z, \theta) \text{ subject to } g(z, \theta) = 0,$$

where $z \in \mathbb{R}^n$ are optimization variables, $\theta \in \mathbb{R}^p$ are parameters, $f : \mathbb{R}^n \times \mathbb{R}^p \to \mathbb{R}$ is a cost function, and $g : \mathbb{R}^n \times \mathbb{R}^p \to \mathbb{R}^q$ defines equality constraints. The solution $z = z(\theta)$ and its associated Karush-Kuhn-Tucker (KKT) multipliers $\lambda = \lambda(\theta) \in \mathbb{R}^q$ must satisfy the KKT conditions

$$F(z, \lambda, \theta) := \begin{bmatrix} \nabla_z L(z, \lambda, \theta) \\ \nabla_\lambda L(z, \lambda, \theta) \end{bmatrix} = \begin{bmatrix} \nabla_z f(z, \theta) + \lambda^\top \nabla_z g(z, \theta) \\ g(z, \theta) \end{bmatrix} = 0,$$

(1)

where $L(z, \lambda, \theta) := f(z, \theta) + \lambda^\top g(z, \theta)$ is the Lagrangian of $\mathbf{P}$.

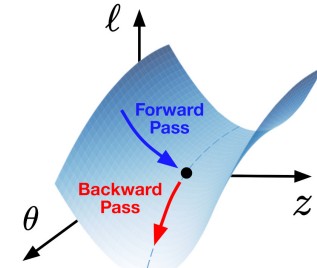

Figure 1: DO solves optimization problems in $z$ (the forward pass) and computes sensitivities with respect to parameters $\theta$ (the backward pass), moving along the loss surface $\ell$ defined by the optimization problem.

The **forward pass** consists of solving the optimization problem $\mathbf{P}$. Many solvers leverage the structure of $\mathbf{P}$ and its KKT conditions, such as its sparsity, to enable efficient numerical resolution. In the context of optimal control, for instance, the iLQR method leverages the sparse-in-time structure of **OCP** and uses Riccati recursions to break the optimization problem into smaller one-step problems that are solved recursively over time $t = 0, \ldots, T$ (Amos et al., 2018). These operations are, however, iterative and do not fully leverage the parallelism of GPUs.

The **backward pass** computes the sensitivities of the solution to $\mathbf{P}$ with respect to the parameters $\theta$. For efficiency, many methods use the implicit function theorem (IFT) for this sensitivity computation (Gould et al., 2016; Amos & Kolter, 2017): By defining the primal-dual pair $w=(z, \lambda)$ solving $\mathbf{P}$,

$$\frac{\mathrm{d}F}{\mathrm{d}\theta} = \frac{\partial F}{\partial w}\frac{\partial w}{\partial \theta} + \frac{\partial F}{\partial \theta} = 0 \implies \frac{\partial w}{\partial \theta} = -\left[\frac{\partial F}{\partial w}\right]^{-1}\frac{\partial F}{\partial \theta}, \quad \text{where} \quad \frac{\partial F}{\partial w} = \begin{bmatrix} \nabla_{zz}L & \nabla_z g^\top \\ \nabla_z g & 0 \end{bmatrix}, \quad (2)$$

where we used $F(z, \lambda, \theta) = 0$ so that $\frac{\mathrm{d}F}{\mathrm{d}\theta} = 0$ in the first equality. The invertibility of the KKT matrix $\frac{\partial F}{\partial w}$ follows from the IFT under suitable assumptions (Lee, 2012; Blondel & Roulet, 2024). Thus, computing the sensitivity matrix $\frac{\partial w}{\partial \theta}$ requires solving $p$ linear systems $\frac{\partial F}{\partial w}\frac{\partial w}{\partial \theta_i} = -\frac{\partial F}{\partial \theta_i}$, which enables computing Jacobian-vector products (JVP) for downstream uses. In machine learning applications, one typically uses the gradient of a function $\ell : \mathbb{R}^n \to \mathbb{R}$ of the solution $z$ to $\mathbf{P}$. Such gradients can be more efficiently computed using the vector-Jacobian product (VJP)

$$\frac{\partial \ell}{\partial \theta}^\top = \frac{\partial w}{\partial \theta}^\top \frac{\partial \ell}{\partial w}^\top = \frac{\partial w}{\partial \theta}^\top \begin{bmatrix} \frac{\partial \ell}{\partial z} \\ 0 \end{bmatrix} = -\frac{\partial F}{\partial \theta}^\top \left(\left[\frac{\partial F}{\partial w}\right]^{-1}\begin{bmatrix} \frac{\partial \ell}{\partial z} \\ 0 \end{bmatrix}\right). \quad (3)$$

Computing the VJP in (3) only requires solving one linear system $\frac{\partial F}{\partial w}\xi = (\frac{\partial \ell}{\partial z}, 0)$, and is thus more efficient than using a JVP, see Appendix A.1 for further details.

Efficient differentiable solvers leverage the structure of $\mathbf{P}$ to quickly compute solutions $z$, and exploit the sparsity structure of the KKT matrix $\frac{\partial F}{\partial w}$ to solve the linear system $\frac{\partial F}{\partial w}\xi = (\frac{\partial \ell}{\partial z}, 0)$ to evaluate the VJPs in (3). In the next section, we describe such a solver for optimal control problems that is designed to leverage parallelism to run efficiently on the GPU.

## 3 DIFFERENTIABLE MODEL PREDICTIVE CONTROL ON THE GPU

`DiffMPC` enables solving and differentiating through optimal control problems (OCPs) of the form

$$\mathbf{OCP}: \quad \arg\min_{z=(x,u)} \sum_{t=0}^{T} c_t^{x,\theta}(x_t) + \sum_{t=0}^{T-1} c_t^{u,\theta}(u_t) \text{ s.t. } f_t^\theta(x_{t+1}, x_t, u_t) = 0, \quad x_0 = x_s^\theta,$$

with separable costs $c_t^{x,\theta}, c_t^{u,\theta}$, equality constraints $f_t^\theta$, and initial conditions $x_s^\theta$ parametrized by $\theta$. In the next sections, we describe its forward pass (solving **OCP**), its backward pass (computing gradients with respect to $\theta$), and discuss design choices for efficient deployment on the GPU.

### 3.1 FORWARD PASS: SOLVING OCP VIA SEQUENTIAL QUADRATIC PROGRAMMING (SQP)

To solve **OCP**, which is in general non-convex, we use the sequential quadratic programming (SQP) scheme with a line search in Algorithm 1. At each iteration of the SQP scheme, given an initial guess for the solution $z$, we approximate the cost of **OCP** by a linear-quadratic function and linearize the dynamics constraints, to obtain the parametric quadratic program (**QP**):

$$\textbf{QP}: \min_{z=(x,u)} \sum_{t=0}^{T} \tfrac{1}{2} x_t^\top Q_t x_t + q_t^\top x_t + \sum_{t=0}^{T-1} \tfrac{1}{2} u_t^\top R_t u_t + r_t^\top u_t \ \text{ s.t. } \ A_t^+ x_{t+1} + A_t x_t + B_t u_t = C_t, \ x_0 = x_s,$$

where $z = (x_0, u_0, \ldots, x_{T-1}, u_{T-1}, x_T)$, and $(Q, q, R, r, A^+, A, B, C)$ depend on the parameters $\theta$, and are computed in parallel over **OCP** problem instances and time steps $t = 0, \ldots, T$:

$$\left(Q_t, q_t\right) := \left(\nabla^2 c_t^{x,\theta}(x_t), \nabla c_t^{x,\theta}(x_t)\right), \quad \left(R_t, r_t\right) := \left(\nabla^2 c_t^{u,\theta}(u_t), \nabla c_t^{u,\theta}(u_t)\right), \tag{4a}$$

$$\left(A_t^+, A_t, B_t\right) := \left(\nabla_{x_{t+1}} f_t^\theta(x_{t+1}, x_t, u_t), \nabla_{x_t} f_t^\theta(x_{t+1}, x_t, u_t), \nabla_{u_t} f_t^\theta(x_{t+1}, x_t, u_t)\right), \tag{4b}$$

and $C_t := A_t^+ x_{t+1} + A_t x_t + B_t u_t - f_t^\theta(x_{t+1}, x_t, u_t)$. To ensure that the matrices $(Q, R)$ are positive definite, we project them onto the positive definite cone as in (Singh et al., 2022). Since the cost is linear-quadratic and the equality constraints are linear, the KKT matrix of **QP** is

$$\frac{\partial F}{\partial w} = \begin{bmatrix} G & H^\top \\ H & 0 \end{bmatrix}, \text{ where } G = \begin{bmatrix} Q_0 & & & \\ & R_0 & & \\ & & \ddots & \\ & & & Q_T \end{bmatrix} \text{ and } H = \begin{bmatrix} I & & & \\ A_0 & B_0 & A_0^+ & \\ & \ddots & \ddots & \ddots \\ & & A_{T-1} & B_{T-1} & A_{T-1}^+ \end{bmatrix}. \tag{5}$$

Solving **QP** consists of finding a pair $(z, \lambda)$ satisfying the KKT conditions in (1), and takes the form:

$$\begin{bmatrix} z \\ \lambda \end{bmatrix} = \begin{bmatrix} G & H^\top \\ H & 0 \end{bmatrix}^{-1} \begin{bmatrix} -b \\ d \end{bmatrix}, \tag{6}$$

where $b = (q_0, r_0, q_1, r_1, \ldots, q_N)$ and $d = (x_s, C_0, \ldots, C_{T-1})$.

### 3.2 BACKWARD PASS: COMPUTING SENSITIVITIES VIA THE IMPLICIT FUNCTION THEOREM

Given a solution $z$ to **OCP** computed via SQP, `DiffMPC` computes the sensitivities with respect to $\theta$ using (3) (see Algorithm 2). The most expensive step consists of solving the linear system

$$\begin{bmatrix} \tilde{z} \\ \tilde{\lambda} \end{bmatrix} = \begin{bmatrix} G & H^\top \\ H & 0 \end{bmatrix}^{-1} \begin{bmatrix} \frac{\partial \ell}{\partial z}(z) \\ 0 \end{bmatrix}. \tag{7}$$

This linear system is the same as the linear system in (6) used for the forward pass, with $(-b, d)$ replaced with $(\frac{\partial \ell}{\partial z}, 0)$. Importantly, the KKT matrix is pre-computed in the forward pass.

### 3.3 SOLVING THE LINEAR SYSTEMS WHILE LEVERAGING PARALLELISM

The efficiency of `DiffMPC` relies on an efficient and GPU-friendly routine for solving the linear systems (6) and (7) that leverages the structure of the KKT matrix in (5). Specifically, `DiffMPC` uses the preconditioned conjugate gradient (PCG) method with tridiagonal preconditioning (Bu & Plancher, 2024) introduced in (Adabag et al., 2024). We briefly describe this method below.

To solve the linear system (6), we first form the Schur complement of the KKT system

$$S := -HG^{-1}H^\top, \qquad \gamma := d + HG^{-1}b, \tag{8}$$

and solve for $\lambda$ and $z$ sequentially:

$$S\lambda = \gamma, \qquad z = -G^{-1}(b + H^\top \lambda). \tag{9}$$

Solving (7) is done similarly, by replacing $(-b, d)$ with $(\frac{\partial \ell}{\partial z}, 0)$. When solving the smaller system in (9), we exploit the structure of $S$ and use a symmetric stair preconditioner (Bu & Plancher, 2024):

$$S = -\begin{bmatrix} Q_0^{-1} & \phi_0^\top & & \\ \phi_0 & \chi_0 & \phi_1^\top & \\ & & \ddots & \phi_{N-2} & \chi_{N-2} & \phi_{T-1}^\top \\ & & & \phi_{T-1} & \chi_{T-1} \end{bmatrix}, \quad \Phi^{-1} := \begin{bmatrix} Q_0 & -Q_0\phi_0^\top\chi_0^{-1} & & \\ -\chi_0^{-1}\phi_0 Q_0 & \chi_0^{-1} & -\chi_0^{-1}\phi_1^\top\chi_1^{-1} & \\ & -\chi_1^{-1}\phi_1\chi_0^{-1} & \chi_1^{-1} & \\ & & & \ddots \end{bmatrix}, \tag{10}$$

where $\chi_t = A_t Q_t^{-1} A_t^\top + B_t R_t^{-1} B_t^\top + A_t^+ Q_{t+1}^{-1} A_t^{+\top}$ and $\phi_t = A_t Q_t^{-1} A_{t-1}^{+\top}$ with $A_{-1}^+ = I$. The solution to (9) is then computed using the PCG method, summarized in Appendix A.2 (see Algorithm 3). Similarly, for the second step in (9), using the structure present in $(G, H)$, the variables $z_t = (x_t, u_t)$ are computed in parallel over $t = 0, \ldots, T$ to maximize efficiency on the GPU:

$$x_t = -Q_t^{-1}\left(q_t + A_{t-1}^{+\top}\lambda_t + A_t^\top \lambda_{t+1}\right), \quad u_t = -R_t^{-1}\left(r_t + B_t^\top \lambda_{t+1}\right), \tag{11}$$

for all $t = 0, \ldots, T-1$, with $A_{-1}^+ = I$ with the last state given by $x_T = -Q_T^{-1}\left(q_T + A_{T-1}^{+\top}\lambda_T\right)$.

While PCG is an iterative routine, we found that it is particularly suitable for differentiable optimization on the GPU as 1) the preconditioner $\Phi^{-1}$ reduces the condition number of $S$ while retaining the parallel-friendly block-tridiagonal structure of $S$, thus enabling parallelization, and 2) its warm-starting capabilities, which are useful when repeating calls to `DiffMPC` in an MPC setting.

### 3.4 Additional Implementation Details and Properties of DiffMPC

Next, we describe details and design choices of `DiffMPC` that optimize for speed and parallelism.

**Line search**: After solving **QP**, a standard line search (Nocedal & Wright, 2006, Algorithm 18.3) is used to select an appropriate step towards the solution of **OCP**. The merit function is defined as a weighted sum of the cost and constraints, and is evaluated in parallel over different predefined step sizes to further leverage GPU parallelism. Details about the line search are in Appendix A.3.

**Warm-starting and reusing computations**: The forward and backward passes of `DiffMPC` can be warm-started with previously computed solutions to the SQP and PCG loops. Also, since the KKT matrix for the forward pass is the same for the backward pass, multiple matrices from the forward pass are passed to the backward pass instead of being recomputed. Figure 2 summarizes data flows.

**Exact SQP vs iLQR**: To form **QP**, an exact SQP scheme would use cost matrices corresponding to the Hessian of the Lagrangian of **OCP** (Nocedal & Wright, 2006) $(Q, R) := \nabla_z^2 L = \nabla_z^2 c + \nabla_z^2 \lambda^\top f$, which requires using the KKT multipliers associated with the equality constraints and additional modifications (e.g., Gauss-Newton approximations) to ensure reliable descent on the problem and that $(Q, R)$ are positive definite. Similarly, the constraints curvature could be accounted for in the backward pass, leading to a different KKT matrix (see, e.g., Frey et al. (2025)). As in (Amos et al., 2018) (and as is standard practice in SQP (Jordana et al., 2025) and done in iLQR), we neglect the curvature of the dynamics to formulate the cost matrices of **QP** as $(Q, R) := \nabla_z^2 c$ and rely on a line search for robustness. This scheme may result in degraded accuracy for the gradients (see e.g. (Frey et al., 2025, Figure 2)). However, it is easier to implement, works well in many applications, and does not require computing second-order derivatives of the constraints that can be computationally expensive to evaluate.

**Parallelism, `DiffMPC` vs iLQR**: Classical algorithms for solving **OCP** such as iLQR use Riccati recursions, and thus operate sequentially over time $t = 0, \ldots, T$ to solve **OCP**. In contrast, `DiffMPC` benefits from multiple sources of parallelism over time steps. First, all matrices are evaluated in parallel for each SQP iteration (e.g., $(Q_t, R_t, A_t, \ldots)$ and blocks of $(S, \Phi^{-1})$). Second, while the PCG routine (Algorithm 3) is iterative, it leverages parallelization over time $t$ for both the forward and backward passes. The warm-starting capabilities of PCG enable fast numerical resolution in MPC applications, whereas the Riccati recursions of iLQR do not leverage warm-starting over problem instances. Leveraging parallelism, warm-starting capabilities, and batching over problem instances makes `DiffMPC` well-suited for learning policies on the GPU.

### 3.5 Reinforcement Learning and Imitation Learning

`DiffMPC` is a fully differentiable policy, making it compatible with standard learning paradigms such as **reinforcement learning** (RL) and **imitation learning** (IL):

$$\textbf{RL:} \quad \max_\theta \mathbb{E}\left[\sum_{t=1}^H R(x_t, \pi^\theta(x_t))\right], \quad x_{t+1} = \text{SimEnv}(x_t, \pi^\theta(x_t)), \quad x_0 \sim \mathcal{D}_{\text{initial states}} \tag{12}$$

$$\textbf{IL:} \quad \min_\theta \mathbb{E}\left[\|(\hat{u}_0, \ldots, \hat{u}_T) - \pi_{0:T}^\theta(x_0)\|^2\right], \quad (x_0, \hat{u}_0, \ldots, \hat{u}_T) \sim \mathcal{D}_{\text{demonstrations}} \tag{13}$$

with an MPC policy $\pi^\theta$ parametrized in $\theta$:

$$\textbf{MPC Policy:} \quad \pi_{0:T}^\theta(x_0) := (u_0^\theta, \ldots, u_T^\theta), \quad \text{where } (x_0^\theta, u_0^\theta, \ldots, x_T^\theta) \text{ solves OCP.} \tag{14}$$

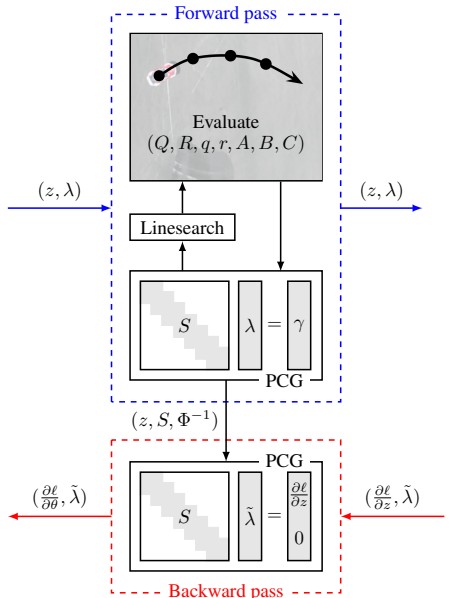

**Algorithm 1** Forward Pass (SQP).

**Inputs**: Initial guess $(x, u, \lambda)$, Tolerance $\epsilon$

1: **while** not converged **do**
2:     Evaluate $(Q, R, q, r, A, B, C)$      Eq. 4
3:     Evaluate $(S, \gamma, \Phi^{-1})$      Eq. 8
4:     $\lambda \leftarrow$ **PCG**$(S, \gamma, \Phi^{-1}, \lambda)$      Alg. 3
5:     Evaluate **QP** solution $(x^+, u^+)$      Eq. 11
6:     $(x, u) \leftarrow$ Linesearch$(x^+, u^+, x, u)$   Sec. A.3
7: **Return**:   Solution $(x, u, \lambda)$, **QP** matrices $(Q, R, \dots)$, Schur matrices $(S, \Phi^{-1})$

**Algorithm 2** Backward Pass (sensitivities).

**Inputs**: Forward pass solution $z = (x, u)$ and matrices $(S, \Phi^{-1})$, Loss gradient $\frac{\partial \ell}{\partial z}$, Initial guess $\tilde{\lambda}$

1: Evaluate $\tilde{\gamma}$ using $(-b, d) \leftarrow (\frac{\partial \ell}{\partial z}, 0)$      Eq. 8
2: $\widetilde{\lambda} \leftarrow$ **PCG**$(S, \widetilde{\gamma}, \Phi^{-1}, \tilde{\lambda})$      Alg. 3
3: Evaluate $\tilde{z}$ using $\tilde{\lambda}$      Eq. 11
4: Evaluate $\frac{\partial \ell}{\partial \theta} \leftarrow -\frac{\partial F}{\partial \theta}^\top \begin{bmatrix} \tilde{z} \\ \tilde{\lambda} \end{bmatrix}$      Eq. 3
5: **Return**: Gradient $\frac{\partial \ell}{\partial \theta}$, solution $\tilde{\lambda}$

Figure 2: `DiffMPC` architecture: forward and backward passes, data flows, and main steps.

In (12), $R$ is a reward function, SimEnv is a simulator for the environment, and $\mathcal{D}_{\text{initial states}}$ is a distribution over initial states. In (13), $\mathcal{D}_{\text{demonstrations}}$ provides demonstration samples for imitation learning. In this work, we conduct RL using a differentiable simulation environment, though this is not required as `DiffMPC` could be used as a component of other differentiable policy architectures. Compared to black-box policies, `DiffMPC` can leverage physics-informed inductive biases through its dynamics model and through solving **OCP**. Its gradients can be computed as described in the previous section. Since the algorithm is tailored for GPUs, large batch sizes can be used for training.

## 4   RESULTS: FASTER SOLVES AND LEARNING ON THE GPU

We implement `DiffMPC` in `JAX` and evaluate it on reinforcement learning and imitation learning tasks. We compare it with three state-of-the-art differentiable solvers: the PyTorch-based nonlinear least-squares solver `Theseus` (Pineda et al., 2022), the PyTorch-based iLQR solver `mpc.pytorch` (Amos et al., 2018), and the JAX-based iLQR solver `Trajax` (Frostig et al., 2021).

### 4.1   TIMING RESULTS FOR REINFORCEMENT LEARNING

We consider randomly-generated MPC problems with quadratic costs and affine dynamics constraints. Details about problem randomization are in the appendix. Since the problems are convex, we disable the line search for all methods and restrict all solvers to a single iteration, which enables a fairer comparison. The RL task consists of maximizing the reward $R(x, u) := -(\|x\|_2^2 + \|u\|_2^2)$ aggregated over 50 environment time steps for a batch size of 64 randomized environments, by learning the MPC's quadratic cost parameters. Forward- and backward-pass computation times measure the time to compute the aggregate reward over batched rollouts and their gradients with respect to the cost parameters. Statistics for each problem are averaged over 10 seeds. Details are in Appendix B.1 along with additional results on other problems.

**Comparison of computation times.** Figure 3 compares solve times of the different solvers. First, for this problem (and all tested, see Table 3 in the appendix), `Theseus` is the slowest, exceeding 80 sec when run on the CPU. This slowdown is likely due to not sufficiently exploiting the time-induced sparse structure of **OCP**. Second, `mpc.pytorch` and `trajax` are not significantly faster on the GPU than on the CPU, which can be attributed to their design based on sequential-in-time Riccati recursions. Third, on the CPU, `DiffMPC` lies in-between `mpc.pytorch` and `trajax`, so `trajax` should be preferred on the CPU as sequential Riccati recursions are better suited for CPU execution. However, on the GPU, `DiffMPC` is significantly faster than other solvers, with a

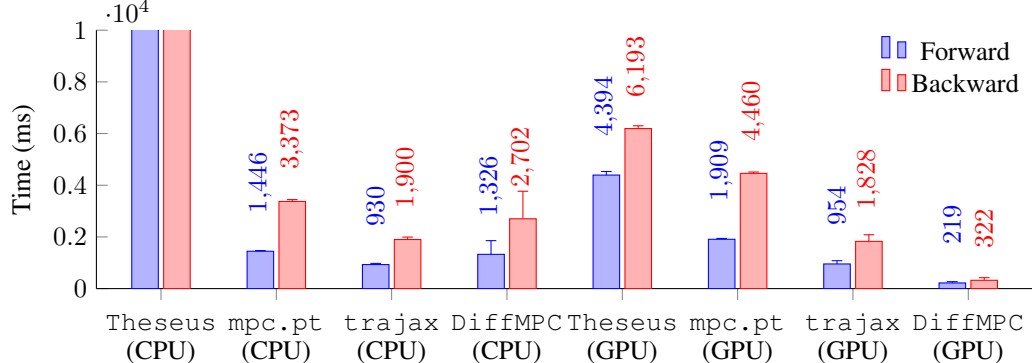

Figure 3: RL computation times on one of the test problems. Error bars indicate $2\sigma$ confidence intervals. Each backward pass also includes one forward pass (to evaluate the inputs of Algorithm 2).

4 times speedup over the fastest baseline for this problem. This speedup is likely due to `DiffMPC` better leveraging parallelism over time in **OCP**. In Appendix B.1, we provide additional results on other problems (including a nonlinear attitude stabilization task), where we also observe significant speedups ranging from 4-7 times over `trajax` (the fastest baseline) across all tested problems.

**Warm-starting.** Using `DiffMPC`'s PCG routine for solving the KKT systems enables warm-starting both the forward and backward passes. Figure 10 in the appendix reports speedups from warm-starting, computed as the ratio $\frac{\text{cold}-\text{warm}}{\text{cold}}$ comparing the RL computation times using `DiffMPC` and of `DiffMPC` with zero initial guesses provided to PCG. For the PCG exit tolerance $\epsilon = 10^{-12}$ that is used in this section's results, warm-starting gives modest speedups of $4\%$ for both the forward and backward passes. These speedups increase to $11\%$ and $9\%$ for the forward and backward passes, respectively, if the tolerance is set to $\epsilon = 10^{-4}$. Thus, we expect additional speedups for low tolerances and in applications where `DiffMPC` is used to replan at high frequencies.

### 4.2 TIMING RESULTS FOR IMITATION LEARNING

Next, we evaluate the method on an imitation learning task with nonlinear dynamics. Following the setup in (Amos et al., 2018), we use a cart-pole environment and cost parameters corresponding to expert imitation data collected by solving **OCP**. To this end, we minimize a standard mean-square-error imitation learning loss, as defined in (13). The goal of this experiment is to evaluate end-to-end training speed on an imitation learning task, with each training loop consisting of solving a batch of nonlinear optimization problems, computing the imitation loss, backpropagating gradients, and updating weights. Details are in Appendix B.2.

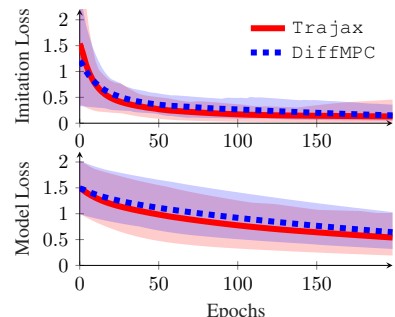

Figure 4: Losses over 200 epochs for the pendulum cart-pole IL benchmark.

On the GPU, we compare `DiffMPC` against `trajax`, as it is the fastest baseline from the previous section. Figure 4 reports the training loss and model loss $\|\theta - \theta^\star\|_2$, where $\theta^\star$ are the true parameters of the policy used to generate the data. `DiffMPC` trains approximately **2× faster** in wall-clock time while maintaining convergence, highlighting its efficiency for imitation learning.

## 5 APPLICATION TO DOMAIN RANDOMIZATION FOR DRIVING AT THE LIMITS

Finally, we demonstrate the practical utility of `DiffMPC` in learning robust controllers for driving at the limits of handling under model mismatch. Existing methods for drifting remain sensitive to modeling errors, as unstable dynamics can cause small errors to quickly amplify and destabilize the vehicle. While prior works have developed learning-based and adaptive controllers (Davydov et al., 2025; Djeumou et al., 2024), online adaptation alone may fail to recover control of a vehicle driving through varying road conditions due to limited actuation. Developing controllers that are robust to

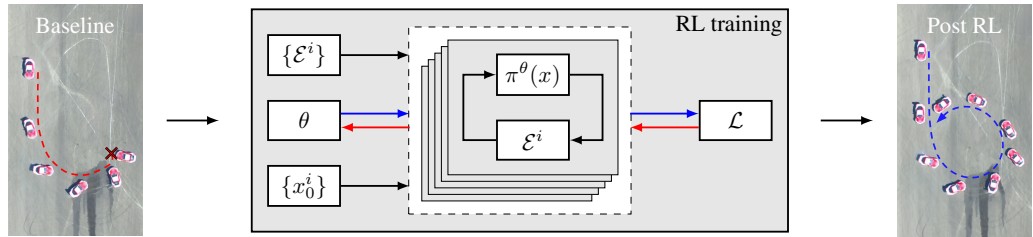

Figure 5: Proposed RL training pipeline to robustify an MPC policy $\pi^\theta(x)$ for drifting.

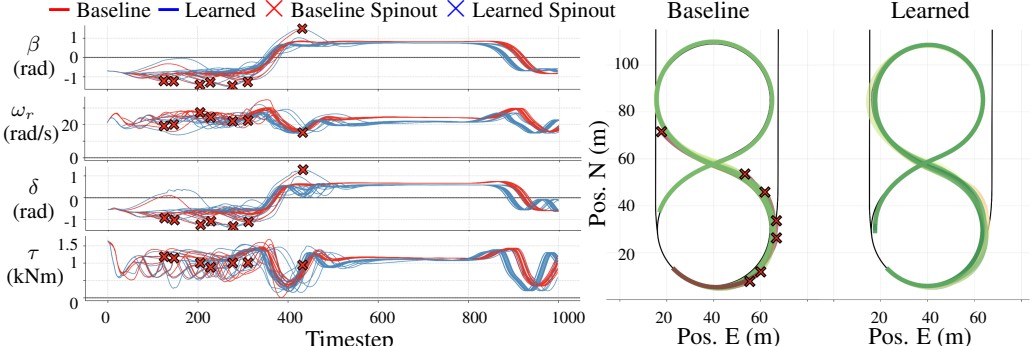

Figure 6: Vehicle states (left) and position trajectories (right) when drifting a figure 8 with puddles.

disturbances such as sudden friction loss from water puddles, and simplifying their tuning process that can be slow and expensive, is crucial for enabling safety-critical applications.

To this end, we use `DiffMPC` within a model-based RL framework for the task of robustly drifting trajectories using domain randomization. Here, we use `DiffMPC` with a differentiable simulator to optimize the MPC problem parameters through backpropagation. We vary the simulator model $\mathcal{E}^i$ by adding water puddles on the road at random locations that reduce available tire forces by modifying physical parameters such as tire friction coefficients, and starting the simulations from varying starting conditions $x_0^i$. For each initial state-environment tuple $(x_0^i, \mathcal{E}^i)$, we generate closed-loop rollouts of 200 steps by repeatedly solving the MPC problem, applying the control $u_t^\theta$, and simulating the evolution of the system given the sampled environments $\mathcal{E}^i$. These simulation environments use high-accuracy dynamics integrators and account for control delays, which would be difficult to do in the MPC controller without increasing the complexity of **OCP**. The total reward is aggregated over the batched rollouts, then backpropagated to learn the policy parameters $\theta$, which consist of the cost weights and the tire friction parameters in **OCP**. We found that using a large episode length ($> 100$ time steps) and batch size ($\geq 32$) is necessary for robust training. We train the policy for 1000 steps with a batch size of 32, taking 14 hours on an NVIDIA GeForce RTX 4090. Further details are in Appendix B.3.

**Changes in parameters after training**: MPC parameters pre- and post-training are in Figure 12 in the appendix. Baseline weights were manually tuned using domain knowledge and tests on a vehicle in nominal dry conditions. The learned policy has significantly decreased the rear tire friction coefficient in its prediction model (change of -13%) and decreased the cost term associated with sideslip angle errors in the objective function of **OCP** (change of -58%). These learned parameters are physically reasonable, but they would have been difficult to obtain by hand, given the surprisingly asymmetric reduction in rear tire friction coefficients. Using these parameters enables the policy to trade off higher sideslip tracking errors for increased robustness to water puddles, and selecting lower engine torques, resulting in more robust drifting as shown next.

**Improved robustness in simulation**: Figure 6 shows twenty roll-outs over randomized environments for the baseline and learned policies, as described in Appendix B.3. The learned policy is significantly more robust, succeeding in 100% of the trials compared to the 70% success rate of the baseline policy. The learned policy selects smaller steering angles and motor torques than the baseline. These actions result in drifting with lower sideslip (the angle between the longitudinal axis and the velocity vector of the vehicle) and lower wheel speed, which gives additional buffers to avoid

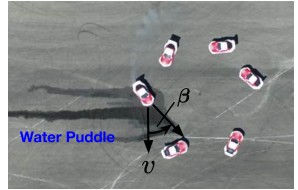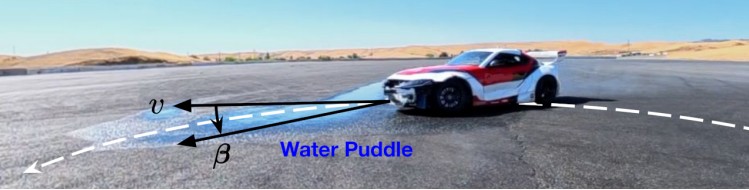

Figure 7: Vehicle drifting through a water puddle: top view (left) and center view (right). The Supra robustly drifts despite variations in friction, which requires carefully selecting actions and sideslip.

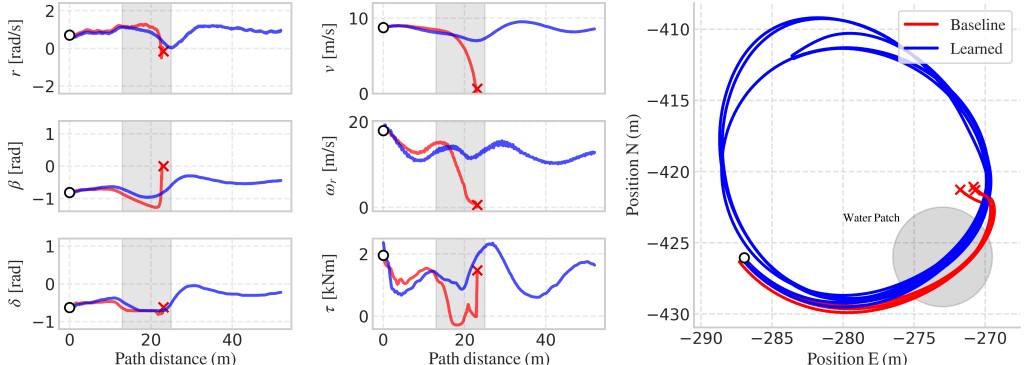

Figure 8: Drifting a donut with a water puddle with a Toyota Supra. Left: State trajectories of two runs with the baseline vs the learned policy. Right: top view trajectories of the runs.

saturating actuation limits and spinning out after drifting through water puddles. These changes result in significant robustness gains despite model mismatch.

**Results on a Toyota Supra**: The learned MPC policy is then deployed on a Toyota Supra drifting a donut trajectory through water puddles. Results are shown in Figures 7 and 8. Although RL training is only conducted on figure-8 trajectories, the learned policy transfers successfully to drifting a circle without additional tuning, thanks to the inductive biases of MPC. In contrast to the baseline that consistently spins out due to the water puddle, the learned policy applies lower engine torques to reduce wheel speeds and maintain lower controlled sideslip angles $\beta$ throughout the drifting manoeuver. Further results for drifting the figure 8 trajectory are in the appendix. Overall, these results show that training a differentiable MPC policy via RL and domain randomization can produce robust, transferable controllers for driving at the limits of handling.

## 6 DISCUSSION AND LIMITATIONS

While differentiable optimization tools often rely on iterative algorithms (such as gradient descent, SQP, and PCG), exploiting parallelism in the problem structure gives opportunities to leverage GPUs to efficiently solve such optimization problems. In this work, we exploit the time-induced sparsity of optimal control problems to yield an efficient differentiable optimization tool for model predictive control that outperforms existing tools when run on the GPU, even for modest batch sizes. This tool offers the strong inductive biases of model-based control in a package that better scales to the demands of data-driven methods, enabling integration of model-based and learning-based approaches.

**Limitations and future work**: First, additional inequality constraints can be accounted for in **OCP** by penalizing them in the cost, and control bounds can be accounted for in the dynamics (e.g., control bounds are enforced in the simulator for RL in Section 5). Handling such inequality constraints via augmented Lagrangian or interior-point methods (Howell et al., 2022; Bambade et al., 2024; Frey et al., 2025; Zuliani et al., 2025) might lead to more reliable convergence and higher-quality solutions. However, reliably differentiating through such problems remains challenging, since gradients can be discontinuous at the boundary of the constraints. Second, since `DiffMPC` is tailored to the GPU and implemented in `JAX`, it runs slower on the CPU than on the GPU. Rewriting the solver in C/C++ would give speedups over our `JAX` implementation, albeit other approaches using Riccati recursions might outperform `DiffMPC` on the CPU. Fourth, `DiffMPC` does not explicitly support tuning solver hyperparameters such as the maximum number of iterations or the PCG tolerance, though its ability to run in parallel over problem instances on the GPU might help tune such hy-

perparameters. Finally, poor initial guesses for the solutions and parameters of differentiable MPC tools can result in divergence of the solver and hinder the downstream training pipeline, motivating future work towards robust initializations for differentiable optimization pipelines.

**Reproducibility statement.** Code to reproduce results in Section 4 is available at

`https://github.com/ToyotaResearchInstitute/diffmpc`

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

# APPENDIX

## A SOLVER: ADDITIONAL DETAILS

### A.1 VECTOR-JACOBIAN PRODUCTS (VJPS) VS JACOBIAN-VECTOR PRODUCTS (JVPS)

From (3), vector-Jacobian products (VJP) for computing gradients of scalar-valued functions of solutions to optimization problems can be computed as

$$\frac{\partial w}{\partial \theta}^{\top} v = -\frac{\partial F}{\partial \theta}^{\top} \left[\frac{\partial F}{\partial w}\right]^{-1} v = -\frac{\partial F}{\partial \theta}^{\top} \begin{bmatrix} \widetilde{z} \\ \widetilde{\lambda} \end{bmatrix}, \quad \text{where} \quad \begin{bmatrix} \widetilde{z} \\ \widetilde{\lambda} \end{bmatrix} \text{ solves} \quad \underbrace{\frac{\partial F}{\partial w}}_{(n+q)\times(n+q)} \underbrace{\begin{bmatrix} \widetilde{z} \\ \widetilde{\lambda} \end{bmatrix}}_{(n+q)} = \underbrace{v}_{(n+q)},$$

where $\frac{\partial F}{\partial w} \in \mathbb{R}^{(n+q)\times(n+q)}$ is symmetric, $(\widetilde{z}, \widetilde{\lambda}) \in \mathbb{R}^{n+q}$, and $v \in \mathbb{R}^{n+q}$. In contrast, computing JVPs of the form $\frac{\partial w}{\partial \theta} v$ requires computing the full Jacobian $\frac{\partial z}{\partial \theta} \in \mathbb{R}^{n \times p}$, which is the solution to $p$ linear systems. Indeed, by (2), each sensitivity $\frac{\partial z}{\partial \theta_i}$ is obtained by solving

$$\frac{\partial F}{\partial w} \begin{bmatrix} \frac{\partial z}{\partial \theta_i} \\ \frac{\partial \lambda}{\partial \theta_i} \end{bmatrix} = -\frac{\partial F}{\partial \theta_i}.$$

Thus, reverse mode differentiation via VJPs is preferred to forward mode differentiation via JVPs in many applications, such as reinforcement learning and imitation learning.

### A.2 PCG ALGORITHM

The Projected Conjugate Gradient (PCG) algorithm introduced in (Adabag et al., 2024) is used to solve the linear KKT systems in `DiffMPC` (see Algorithm 3). Note that $\gamma$ in (9) takes the form

$$\gamma = d + (Q_0^{-1} q_0, \zeta_0, \zeta_1, \ldots, \zeta_{T-1}),$$

with $\zeta_t = A_t Q_t^{-1} q_t + B_t R_t^{-1} r_t + A_k^{+} Q_{t+1}^{-1} q_{t+1}$ for all $t$. Also, lines 1, 2, 6, and 11 of Algorithm 3 are parallelized over time, reducing computation times when executed on the GPU.

### A.3 LINE SEARCH

After solving each **QP**, we use a standard line search method (Nocedal & Wright, 2006, Algorithm 18.3) described next. Below, we denote by $z^{+} = (x^{+}, u^{+})$ the solution to **QP**, which is evaluated at $z = (x, u)$. The objective of the line search is to select an appropriate step size $\alpha \in (0, 1]$ so the new solution $(x + \alpha(x^{+} - x), u + \alpha(u^{+} - u))$ is closer to an optimal solution to **OCP**.

The line search uses a merit function $\varphi$ defined as the weighted sum of the cost and constraints

$$\varphi(x, u) := \sum_t \left( c_t^x(x_t) + c_t^u(u_t) + \mu \left\| f(x_{t+1}, x_t, u_t) \right\|_1 \right), \tag{15}$$

where the penalty parameter $\mu$ is set as in (Nocedal & Wright, 2006, Equation 18.33 with $\rho = 0.5$):

$$\mu = \frac{\sum_t \nabla c_t^x(x_t)^{\top}(x_t^{+} - x_t) + \nabla c_t^u(u_t)^{\top}(u_t^{+} - u_t)}{0.5 \sum_t \left\| f(x_{t+1}, x_t, u_t) \right\|_1}. \tag{16}$$

The decrease condition for accepting a step size $\alpha$ is (Nocedal & Wright, 2006, Equation 18.28)

$$\Delta\varphi_{\alpha} := \varphi(x + \alpha\Delta x, u + \alpha\Delta u) - \varphi(x, u) - \eta\alpha D_{\mu}. \tag{17}$$

for some $\eta \in (0, 1)$ (we use $\eta = 0.4$), with descent direction defined as (Nocedal & Wright, 2006, Equation 18.29)

$$D_{\mu} = \sum_t \nabla c_t^x(x_t)^{\top}(x_t^{+} - x_t) + \nabla c_t^u(u_t)^{\top}(u_t^{+} - u_t) - \mu\|f(x_{t+1}, x_t, u_t)\|_1. \tag{18}$$

Merit function values $\varphi(x + \alpha\Delta x, u + \alpha\Delta u)$ are evaluated in parallel over different pre-defined step sizes in decreasing order $\overset{\rightarrow}{\alpha} := \{1.0, 0.7, 0.3, 0.1, 0.01\}$. The line search method is described in further details in Algorithm 4.

**Algorithm 3** PCG for solving $S\lambda = \gamma$ in (10) (Adabag et al., 2024).

**Inputs**: $S, \gamma, \Phi^{-1}$, initial guess $\lambda$, tolerance $\epsilon$
1:   $r = \gamma - S\lambda$      $\triangleright r_t = \gamma_t - S_t\lambda_{t-1:t+1}$
2:   $\tilde{r} = \Phi^{-1}r$      $\triangleright \tilde{r}_t = \Phi_t^{-1}r_{t-1:t+1}$
3:   $p = \tilde{r}$
4:   $\eta = r^\top\tilde{r}$
5:   **while** $\eta > \epsilon$ **do**
6:     $y = Sp$      $\triangleright y_t = S_tp_{t-1:t+1}$
7:     $v = p^\top y$
8:     $\alpha = \eta/v$
9:     $\lambda = \lambda + \alpha p$
10:    $r = r - \alpha y$
11:    $\tilde{r} = \Phi^{-1}r$      $\triangleright \tilde{r}_t = \Phi_t^{-1}r_{t-1:t+1}$
12:    $\eta' = r^\top\tilde{r}$
13:    $\beta = \eta'/\eta$
14:    $p = \tilde{r} + \beta p$
15:    $\eta = \eta'$
16: **return**: solution $\lambda$ to the system $S\lambda = \gamma$

**Algorithm 4** Linesearch for SQP.

**Inputs**: Previous solution $z = (x, u)$ for formulating **QP**, solution $z^+ = (x^+, u^+)$ to **QP**, candidate step sizes $\overset{\vee}{\alpha} = \{1.0, 0.7, \dots\}$
1: Evaluate merit function $\varphi(x, u)$      Eq. 15
2: Evaluate penalty parameter $\mu$      Eq. 16
3: Evaluate descent direction $D_\mu$      Eq. 18
4: **for** $\alpha \in \overset{\vee}{\alpha}$ **do**      in parallel
5:    Evaluate decrease condition $\Delta\varphi_\alpha$   Eq. 17
6: $\alpha \leftarrow \min(\overset{\vee}{\alpha})$
7: **for** $\widetilde{\alpha} \in \overset{\vee}{\alpha}$ **do**      sequentially
8:    **if** $\Delta\varphi_{\widetilde{\alpha}} < 0$ **then**
9:      $\alpha \leftarrow \widetilde{\alpha}$      select largest step size
10:    Break
11: **Return**: New solution $(x + \alpha(x^+ - x), u + \alpha(u^+ - u))$

# B   RESULTS: ADDITIONAL DETAILS AND EXPERIMENTS

In this section, $y \sim \mathcal{N}(\bar{y}, \Sigma)$ denotes a Gaussian random variable with mean $\bar{y}$ and covariance matrix $\Sigma$, and $y \sim \mathcal{U}([a, b])$ denotes a uniform random variable in the interval $[a, b]$.

## B.1   REINFORCEMENT LEARNING BENCHMARK

For the RL benchmark, we start by describing the solvers used and how they were modified to produce a fair comparison of `Theseus`, `mpc.pytorch`, `trajax`, and `DiffMPC`. We then consider a linear problem to facilitate ablations on the problem size and study the impact of warm starting on the PCG algorithm. Finally, we consider a nonlinear spacecraft example where we compare the computation times for `trajax` and `DiffMPC`.

**Setup.** For this RL benchmark, we use an evaluation scheme similar to the RL pipeline in Fig. 5. All methods use compute gradients via implicit differentiation and use float64 precision. Timing is conducted on a workstation with a 64-Core AMD Threadripper 3990X CPU and a NVIDIA GeForce RTX 3090 GPU running Ubuntu 22.04. For these evaluations on **QP** problems, `mpc.pytorch` is constrained to 1 iLQR iteration, the line search is disabled, and flags are set so that it exits after the first iteration without detaching gradients. The original code is slightly modified to eliminate a serial

Table 1: Computation times (s) with `mpc.pytorch` for Problem 1 on the GPU.

| Pass | Original | Modified |
|------|----------|----------|
| Fwd | 67.9 | **1.9** |
| Bwd | 136.4 | **4.5** |

list comprehension that causes the library to scale poorly when inequality constraints are disabled. Table 1 compares our modified version of `mpc.pytorch` to the base version on the first seed from problem 1, and shows that our modifications reduce the solve times of the forward pass and gradient computations. Since `Theseus` only supports unconstrained non-linear least-square optimization problems, the dynamics are rolled out in the objective function as done in (Wan et al., 2024). The damped least squares method is used and constrained to 1 iteration of the dense Cholesky solver. `Trajax` uses the in-built iLQR **OCP** solver, and is constrained to 1 iLQR iteration and a minimum line search step of 0.99 (we found that a minimum of 1 prevents any solution from being accepted). The `tvlqr` method is used for the backward pass. `DiffMPC` is also limited to 1 **QP** solve and 1 line search iteration, with $\overset{\vee}{\alpha} = \{1\}$. The underlying PCG solver uses an exit tolerance of $10^{-12}$ for both the forwards and backwards passes.

**QP problems.** We consider the six problems defined in Table 2. The dynamics of each problem are defined as $x_{t+1} = Ax_t + Bu_t + b$. The dynamics matrix $A$ is randomized according

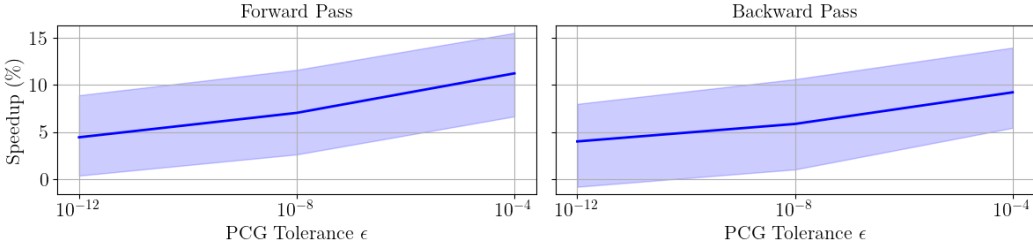

Figure 10: RL with `DiffMPC`: Speedups from warm-starting $\frac{\text{SolveTime}_{\text{cold}} - \text{SolveTime}_{\text{warm}}}{\text{SolveTime}_{\text{cold}}}$ for different PCG exit tolerances $\epsilon$, with $\pm 2$ standard deviation intervals.

to $A = I + 0.1\Delta A$ with $\text{vec}(\Delta A) \sim \mathcal{N}(0, I)$, and its eigenvalues are clipped to lie within the unit circle so that $|\lambda(A)| \in (0, 0.99]$. Similarly, we let $\text{vec}(B) \sim \mathcal{N}(0, I)$, $b \sim \mathcal{N}(0, 0.10^{-4}I)$, and sample initial conditions as $x_0 \sim \mathcal{N}(0, 25I)$. The cost functions in **OCP** are quadratic functions $c_t^{x,\theta}(x) = x^\top Q x$ where $Q$ is a diagonal matrix and $c_t^{u,\theta}(u) = \|u\|_2^2$ for all $t$. The RL reward is defined as $R(x, u) = -(\|x\|_2^2 + \|u\|_2^2)$. The parameters $\theta$ that are optimized are $\theta = \text{diag}(Q)$. For each of these problems, we compute statistics over ten independent runs per method, as we observed little variation in computation times within each problem (apart from `Theseus` on the CPU). Timing results are in Table 3. The results of Problem 1 are illustrated in Figure 3, and are representative of performance across the tested problem instances.

**Warm-starting.** We provide further results in Figure 10 to assess the benefits from warm-starting `DiffMPC` for different PCG exit tolerances $\epsilon \in \{10^{-4}, 10^{-8}, 10^{-12}\}$ on Problem 1. Statistics are computed over 100 independent runs for each PCG exit tolerance. We observe speedups for both the forward and backward passes. Speedups are more significant for lower PCG exit tolerances $\epsilon$. We note that tolerances in the order $\epsilon = 10^{-4}$ are reasonable for many applications, see the results in (Adabag et al., 2024). To ensure fair comparisons

Table 2: Parameters in the RL benchmark: state dimension $n_x$, control dimension $n_u$, MPC horizon $T$, RL episodes length $H$, batch size $B$.

| Problem | $n_x$ | $n_u$ | $T$ | $H$ | $B$ |
|---------|-------|-------|-----|-----|-----|
| 1 | 8 | 4 | 40 | 50 | 64 |
| 2 | 8 | 4 | 30 | 50 | 16 |
| 3 | 8 | 4 | 30 | 50 | 64 |
| 4 | 8 | 4 | 30 | 50 | 256 |
| 5 | 16 | 8 | 30 | 50 | 16 |
| 6 | 16 | 8 | 30 | 50 | 64 |

against baselines, we used the lowest tolerance $\epsilon = 10^{-12}$ in all other results in the paper, and anticipate that additional speedups compared to these baselines could be observed by selecting a smaller exit tolerance $\epsilon$.

**Accuracy of computed gradients.** We evaluate the accuracy of the gradients computed by `DiffMPC` against finite-difference approximations and report results in Figure 9. These statistics are computed over 100 problem instances for each PCG exit tolerance. For this problem, we observe a high degree of consistency between `DiffMPC` and finite differences up to a PCG exit tolerance of $10^{-6}$. We use an exit tolerance of $10^{-12}$ for all other results.

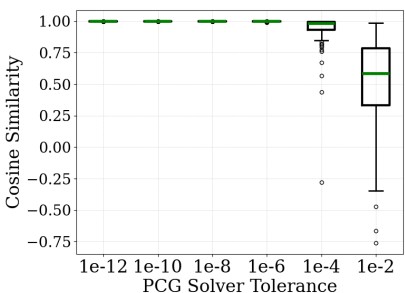

Figure 9: Cosine similarity between computed gradients and finite differencing across PCG exit tolerances.

Table 3: Timing results on CPU and GPU across 6 problem instances with mean time and $2\sigma$ (s).

| Method | GPU | Pass | Problem 1 | Problem 2 | Problem 3 | Problem 4 | Problem 5 | Problem 6 |
|---|---|---|---|---|---|---|---|---|
| Theseus | ✗ | Bwd | 81.47 (34.50) | 5.04 (0.46) | 10.44 (3.28) | 130.01 (2.94) | 9.27 (3.70) | 130.67 (5.73) |
| Theseus | ✗ | Fwd | 49.35 (70.65) | 3.39 (0.18) | 6.10 (2.18) | 65.44 (63.67) | 5.63 (2.45) | 70.74 (85.68) |
| mpc.pytorch | ✗ | Bwd | 3.37 (0.07) | 1.71 (0.03) | 2.60 (0.07) | 5.61 (0.08) | 2.32 (0.04) | 4.01 (0.07) |
| mpc.pytorch | ✗ | Fwd | 1.44 (0.03) | 0.72 (0.01) | 1.12 (0.03) | 2.36 (0.03) | 0.97 (0.01) | 1.65 (0.02) |
| trajax | ✗ | Bwd | 1.90 (0.09) | 0.34 (0.02) | 1.42 (0.07) | 5.95 (0.27) | 0.94 (0.03) | 3.89 (0.10) |
| trajax | ✗ | Fwd | 0.93 (0.05) | 0.17 (0.01) | 0.70 (0.03) | 2.96 (0.15) | 0.45 (0.01) | 1.91 (0.05) |
| DiffMPC | ✗ | Bwd | 2.70 (1.07) | 0.35 (0.10) | 1.49 (0.44) | 10.00 (3.71) | 1.58 (0.30) | 10.96 (2.97) |
| DiffMPC | ✗ | Fwd | 1.33 (0.53) | 0.16 (0.09) | 0.71 (0.23) | 5.09 (1.72) | 0.74 (0.18) | 5.49 (1.38) |
| Theseus | ✓ | Bwd | 6.19 (0.11) | 4.56 (0.14) | 4.63 (0.14) | 6.49 (0.03) | 4.76 (0.10) | 6.39 (0.04) |
| Theseus | ✓ | Fwd | 4.39 (0.14) | 3.24 (0.17) | 3.26 (0.05) | 5.20 (0.02) | 3.41 (0.12) | 5.16 (0.04) |
| mpc.pytorch | ✓ | Bwd | 4.46 (0.05) | 3.33 (0.10) | 3.40 (0.05) | 3.98 (0.04) | 3.33 (0.04) | 3.43 (0.06) |
| mpc.pytorch | ✓ | Fwd | 1.91 (0.03) | 1.45 (0.05) | 1.45 (0.01) | 1.67 (0.02) | 1.45 (0.05) | 1.47 (0.03) |
| trajax | ✓ | Bwd | 1.83 (0.26) | 1.32 (0.19) | 1.36 (0.18) | 3.81 (0.63) | 2.03 (0.04) | 2.20 (0.04) |
| trajax | ✓ | Fwd | 0.95 (0.13) | 0.70 (0.09) | 0.71 (0.91) | 1.93 (0.33) | 1.05 (0.02) | 1.13 (0.02) |
| DiffMPC | ✓ | Bwd | **0.32** (0.10) | **0.18** (0.06) | **0.27** (0.08) | **0.69** (0.16) | **0.32** (0.08) | **0.73** (0.16) |
| DiffMPC | ✓ | Fwd | **0.22** (0.05) | **0.11** (0.03) | **0.18** (0.03) | **0.49** (0.08) | **0.20** (0.04) | **0.45** (0.08) |

**Nonlinear attitude stabilization.** We also consider a nonlinear attitude stabilization task, where we drive the attitude rates of a rigid-body $x = \omega \in \mathbb{R}^3$ (rad/s) to zero by actuating the torques $u = \tau \in \mathbb{R}^3$. The nonlinear continuous-time system dynamics are

$$J\dot{\omega} = J\omega \times \omega + \tau, \qquad (19)$$

where $J \succ 0$ is a diagonal inertia matrix. We discretize the system using a forward Euler scheme with a time step

Table 4: Compute times on the nonlinear attitude stabilization problem.

| Method | Pass | Compute time (ms) |
|---|---|---|
| trajax | Fwd. | 310.6 (24.1) |
| trajax | Bwd. | 505.0 (27.4) |
| DiffMPC | Fwd. | 39.8 (8.0) |
| DiffMPC | Bwd. | 69.3 (16.1) |

of $\Delta t = 0.1$ s. We consider an MPC policy with a prediction horizon $T = 25$ and costs defined as in **QP** with nominal time-invariant cost matrices $Q = R = I$. The learned parameters are defined as $\theta = (\text{diag}(Q), \text{diag}(R))$, and we train these parameters via gradient descent on the RL reward $R(x, u) = -(0.1\|x\|_2^2 + \|u\|_2^2)$ using a batch size of $B = 16$. In each batch, we draw random initial conditions from $\omega_0 \sim \mathcal{U}([-1, 1]^3)$ and randomize the inertia matrix as $\text{diag}(J) \sim \mathcal{U}([0.1, 10]^3)$. We compare the compute times on the backward and forward passes for this problem with trajax and DiffMPC. Results are reported in Table 4. We note a significant speedup (approximately $7.3\times$ for the backward pass) when solving this problem using DiffMPC compared to trajax.

### B.2 IMITATION LEARNING BENCHMARK

For this task, we consider the cart-pole problem in (Amos et al., 2018), with states $x = (x_1, x_2, x_3, x_4) \in \mathbb{R}^4$ (m, m/s, rad, rad/s) and inputs $u \in \mathbb{R}$ (N). The model is parametrized with a length $L$, the cart mass $m_c$, the pole mass $m_l$, and the gravitational constant $g$. Its dynamics are

$$\dot{x}_1 = x_2, \qquad (20a)$$

$$\dot{x}_2 = \frac{-m_p L \sin(x_3)x_4^2 + m_p g \sin(x_3)\cos(x_3)}{(M + m_p(1 - \cos^2(x_3))L}, \qquad (20b)$$

$$\dot{x}_3 = x_4, \qquad (20c)$$

$$\dot{x}_4 = \frac{-m_p L \sin(x_3)x_4 + m_p g \sin(x_3)\cos(x_3) + u}{M + m_p(1 - \cos^2(x_3))}, \qquad (20d)$$

and we let $m_c = 1$, $m_p = 0.1$, $l = 0.5$, $g = 9.81$, with $M = m_p + m_c = 1.1$. To generate expert data, we use DiffMPC in the form of **OCP**, with the quadratic objective in **QP**

$$Q_t = \text{diag}(1, 2, 1.5, 1), \quad \text{and} \quad R_t = 0.05 \quad \forall t = 0, ..., T. \qquad (21)$$

Based on this expert policy, 32 different initial conditions $x_0 = (x_{10}, x_{20}, x_{30}, x_{40})$ are sampled according to the uniform distributions $x_{10} \sim \mathcal{U}([-0.5, 0.5])$, $x_{20} \sim \mathcal{U}([-0.5, 0.5])$, $x_{30} \sim$

Table 5: Nominal vehicle model parameters and controller gains for the baseline MPC policy.

| | $Q_{11}$ | $Q_{22}$ | $Q_{33}$ | $Q_{44}$ | $Q_{55}$ | $Q_{66}$ | $Q_{77}$ | $R_{11}$ | $R_{22}$ | $a$ | $b$ | $m$ | $r_{\mathrm{w}}$ | $\mu_f$ | $\mu_r$ | $C_f$ | $C_r$ | $I_{\mathrm{w}}$ |
|---|---|---|---|---|---|---|---|---|---|---|---|---|---|---|---|---|---|---|
| Value | $10^{-4}$ | $10^{-1}$ | 500 | 1 | 20 | 300 | $10^{-6}$ | 10 | $10^{-4}$ | 1.239 | 1.209 | 1476 | 0.323 | 0.99 | 0.90 | 54000 | 220000 | 11.28 |

$\mathcal{U}([-\pi, \pi])$, and $x_{40} \sim \mathcal{U}([-1, 1])$. From each initial condition, an expert trajectory is generated using `DiffMPC` with the costs in (21). For imitation learning, we learn the cost parameters $\theta = (Q_{11}, Q_{22}, Q_{33}, Q_{44})$ and draw the initial parameters of the learned policy from a uniform distribution $\mathcal{U}([0, 1])$. Training is conducted using a batch size of 32, following Amos et al. (2018), by gradient descent using a learning rate of $10^{-2}$ on the mean-square-error objective defined in (13). Both `trajax` and `DiffMPC` were limited to five SQP and iLQR iterations and are warm-started using the expert trajectory.

### B.3 DRIFTING EXPERIMENTS

**Vehicle model.** For the drifting experiments, we consider a dynamic bicycle model (Lew et al., 2025) for both planning and simulation. We model tire forces using a coupled slip brush Fiala model (Svendenius, 2007). In this setting, the state of the system is defined by $x = (r, v, \beta, \omega_r, \Delta\phi, e, s) \in \mathbb{R}^7$, where $r$ (rad/s) is a yaw rate, $v$ (m/s) is a longitudinal velocity, $\beta$ (rad) is a sideslip angle, $\omega_r$ (rad/s) is the average rear wheel speeds, $\Delta\phi$ (rad) is a heading tracking error with respect to a reference, $e$ is a lateral tracking error to the same reference, and $s$ (m) is the path distance traveled along the reference trajectory. The car is driven by the control input $u = (\delta, \tau)$, where $\delta$ (rad) is the steering angle, and $\tau$ (Nm) is the engine torque. The simulator discretizes the vehicle's dynamics using a high-accuracy Dormand-Prince method implemented in `Diffrax` (Kidger, 2021), whereas the MPC model uses a simple trapezoidal scheme discretized at $\Delta t = 0.1$ sec. Box constraints on the control inputs are enforced in the simulator but omitted in the MPC controller so that robustness to actuator saturation is learned implicitly through training, and so that simpler unconstrained problems can be solved both during training and at runtime.

The vehicle dynamics are parameterized by the parameters $(a, b, I_z, m, r_{\mathrm{w}}, I_{\mathrm{w}}, C_f, C_r, \mu_f, \mu_r) \in \mathbb{R}^{10}_{>0}$, where $(a, b)$ define the location of the center of mass of the vehicle, $m$ is its mass, $r_{\mathrm{w}}$ is the wheel radius, $I_z$ is the moment of inertia about the yaw axis, $I_{\mathrm{w}}$ is the moment of inertia of a wheel, and $C_i$ and $\mu_i$ denote the cornering stiffness and friction coefficients of the front and read wheels $i \in \{f, r\}$, respectively. Nominal vehicle parameters are given in Table 5.

**MPC.** The cost function of the MPC policy is defined with quadratic costs

$$c_t^{x,\theta}(x_t) = \tfrac{1}{2}(x_t - x_{\mathrm{ref},t})^\top Q (x_t - x_{\mathrm{ref},t}), \quad c_t^{u,\theta}(\dot{u}_t) = \tfrac{1}{2}\dot{u}_t R \dot{u}_t, \tag{22}$$

where $c_t^{u,\theta}(\dot{u}_t)$ penalizes control rates $\dot{u}_t = (u_{t+1} - u_t)/\Delta t$ (this cost is implemented in the MPC by augmenting the state with the control input as $(x \leftarrow (x, u))$ and redefining the control input as the control rate $(u \leftarrow \dot{u})$), and $x_{\mathrm{ref}}$ is a reference trajectory computed using the method in (Goh & Gerdes, 2016) with nominal vehicle parameters. Experiments involve drifting along the figure 8 trajectory in Figure 6 and the donut trajectory in Figure 8.

**Learnable parameters.** The learnable parameters $\theta$ of the MPC policy are the cost weights for the states and control rates, the front and rear tire friction values, the front and rear cornering stiffness, and the rear wheels inertia:

$$\theta = (\mathrm{diag}(Q), \mathrm{diag}(R), \mu_f, \mu_r, C_f, C_r, I_{\mathrm{w}}) \in \mathbb{R}^{14}_{>0}. \tag{23}$$

**RL training procedure.** To learn a robust drifting policy, we define the RL reward as

$$R(x) = -\left( (r - r_{\mathrm{ref}})^2 + \lambda e^{-\gamma \beta^2} \right), \tag{24}$$

where $\lambda = 0.3$ and $\gamma = 50$. This reward optimizes for non-zero side-slip angles (to ensure drifting behavior) and prioritizes the heading reference tracking. The RL batch size is set to $B = 32$, and the episode length is set to $H = 200$, with each step taking 20ms. We found that RL training without a term rewarding high sideslips $\beta$ yields policies that drive robustly on the reference path without drifting, highlighting the challenge of controlling a vehicle in an unstable drifting regime. To rescale the learned cost parameters that have different magnitudes, and ensure that they remain

Table 6: Parameters used for domain randomization. On water puddles, tire friction coefficients drop to $\mu = 0.6$.

| | $\mu_{\text{f}}$ | $\mu_{\text{r}}$ | $10^{-3}C_{\text{f}}$ | $10^{-3}C_{\text{r}}$ | $I_{\text{w}}$ | $s_0$ | $\ell_{\text{puddle}}$ | $s_{\text{puddle}}$ |
|---|---|---|---|---|---|---|---|---|
| range | $[0.94, 1.04]$ | $[0.85, 0.95]$ | $[52, 56]$ | $[200, 240]$ | $[8, 14]$ | $[185, 700]$ | $[0, 5]$ | $[-1, 1]$ |

strictly positive, they are log-normalized for training. Training is conducted on the figure 8 trajectory only. The reward in (24) is maximized via gradient ascent with a constant learning rate of $0.1$.

*Randomization*: The parameters for each of the $B = 32$ environments are drawn from uniform distributions defined in Table 6 where $s_0$ represents the initial state along the reference trajectory in meters, $\ell_{\text{puddle}}$ represents the length of the puddle, and $s_{\text{puddle}}$ represents the relative distance between where the front and rear wheels enter the puddle. We note that puddles can be of irregular shapes and drifting involves controlling a vehicle sideways, so front wheels do not necessarily enter water puddles before the rear wheels.

*MPC*: During training, a single SQP iteration is used per time step to match the real-time iteration scheme that is used during deployment (Gros et al., 2020). The MPC uses a horizon of $T = 35$ timesteps. The rollout loop used for simulation accounts for control delays present in the real application by introducing a 20 ms delay to the incoming control signal, corresponding to a conservative estimate of the time needed to replan on the vehicle using MPC.

*Low-level interpolation*: On the vehicle, a low-level interpolation scheme is used to select control inputs from the MPC plan to send to the actuators. This interpolation scheme is also accounted for in simulation for RL. The MPC stack deployed on the vehicle is identical to the stack used in training with `DiffMPC`, except that it uses OSQP (Stellato et al., 2020) as the QP solver instead of the PCG routine.

**Training dynamics and memory requirements**. In Fig. 11, we report the training loss across 3 seeds for the first 100 steps of the drifting MPC controller RL experiment. These results were collected on a workstation with a NVIDIA GeForce RTX 3090 GPU using a batch size of $B = 32$, where it used 18GB of VRAM. These results show that despite noise in the training curves, the loss consistently stabilizes to a minimum after around 40 minutes of training for these runs.

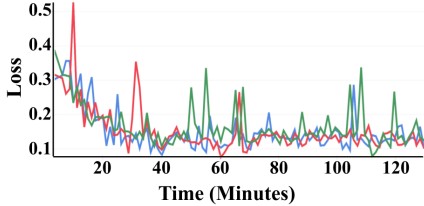

Figure 11: Wall-clock training loss curves across 3 seeds.

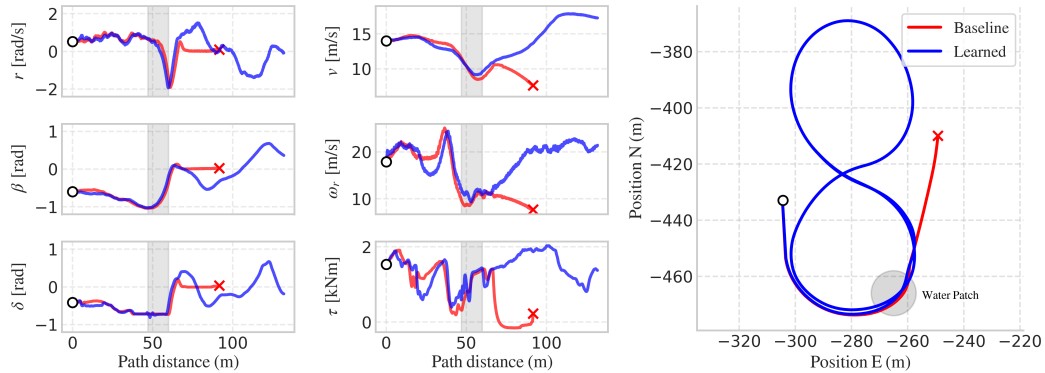

Figure 13: On-vehicle test for drifting a figure 8 trajectory with a water puddle in the first turn. Closed-loop response task with the baseline (red) and the learned controller (blue).

**Additional simulation results.** In addition to the simulation results reported in Section 5, we provide the result of training with puddles in the environment. In Figure 12, we show the relative change in the learned parameters with respect to the nominal parameters in Table 5. The more robust policy produced by the RL training uses lower tire friction values. Using these lower friction parameters has the effect of reducing the torques and wheel speeds as the planner assumes the tires have less grip than they actually do in nominal road conditions, resulting in a more shallow drift. The learned policy also exhibits a 58% reduction in sideslip cost $\beta$. Reducing this cost term on sideslip enables the policy to drift with lower sideslip (which leads to worse tracking error) for increased robustness to potential spinouts.

**Additional hardware results: drifting the figure 8**. For completeness, in Figure 13, we report additional on-vehicle results for drifting the figure 8 trajectory in the presence of a water puddle in the first turn. In these experiments, the baseline spins out in the first turn, whereas

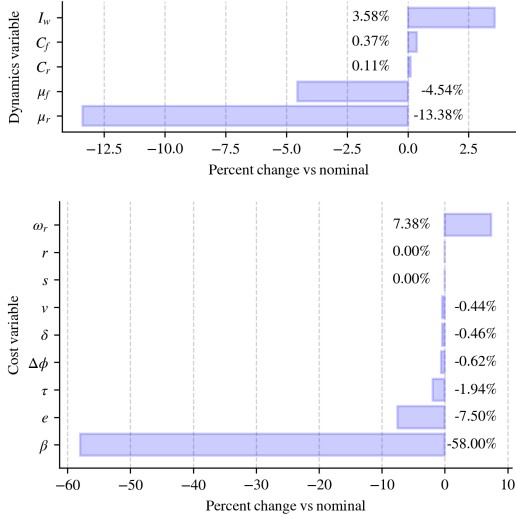

Figure 12: Change in MPC policy parameters after learning $\theta$ by domain randomization that includes water puddles.

the controller using the parameters learned using `DiffMPC` completes the maneuver. However, we were unfortunately only able to run this experiment once for each method due to hardware limitations. For this reason, we refrain from drawing strong conclusions from these results.

