# OpenReview forum: "Differentiable Model Predictive Control on the GPU"
_ICLR.cc/2026/Conference — ICLR 2026 Oral_

### Official Review · Reviewer_4NTm · 2025-10-31

**Soundness:** 3
**Presentation:** 4
**Contribution:** 3
**Rating:** 8
**Confidence:** 3

**Summary:**

The paper presents a solver for Model Predictive Control (MPC) which is optimized for running on a GPU. The optimization tool called DiffMPC is differentiable and can be integrated in RL or imitation learning tasks. The method uses line search, warm start of both forward and backward pass, and parallelization over time steps to improve efficiency on the GPU. Experiments show that it runs faster on the GPU than other MPC solvers, and can be used effectively for RL and imitation learning tasks, including a simulation of driving in new domains.

**Strengths:**

1. The method is tested in randomized environments as well as real applications, and the results in Figure 3 and Table 3 in the appendix show that it consistently outperforms the other methods.
2. There are some ablations of individual aspects of the method, such as the warm start, which is convincing.
3. The paper motivates the approach well and puts it into context with prior work, pointing out the shortcomings of GPU acceleration. It provides a good background section on differentiable control. It is convincing that this method can improve the integration of learning and optimization.
4. The limitations are described transparently.

**Weaknesses:**

1. One limitation seems to be that only equality constraints are supported. This seems to impact comparability to the baselines (e.g. the appendix states that mpc.pytorch performance suffers when inequality constraints are disabled, which was tackled with some modifications to the code.)
2. Only the synthetic randomized experiments compare the performance between all optimizers. In the imitation learning experiment, only trajax is compared, and in the driving-application, no other optimizer is tested.

**Questions:**

1. How much do the solver outputs (the optimized action u etc) in the RL experiment differ? I.e., does the DiffMPC solver achieve the same performance in one iteration as other solvers, or can the modifications lead to suboptimal results?
2. In Figure 4, it is not visible that there is a 2x speedup; the loss of DiffMPC over time looks the same as trajax. Where is the 2x speedup shown?
3. Why is line search disabled in the RL experiment? Is it not needed when the problem is convex? It seems to be a core part of the method. Is it used for imitation learning or the other experiment?

---

> ### Author Response · Authors · 2025-11-24
>
> Thank you for the constructive and thoughtful feedback. We hope that the following responses address your concerns.
> 1. **Solver Accuracy in RL Experiment (Question 1)**: We checked that the solver outputs in the RL experiment are all the same up to a tolerance of 1e-4. Regarding gradient computation, we have added a figure in Appendix B reporting the accuracy of the gradients against finite differencing, showing that the gradients computed by DiffMPC are highly accurate for this problem.
> 2. **Unclear Speedup in Imitation Learning Figure (Question 2)**: The figure previously showed training and imitation losses as a function of wall clock time, stopping at 200 epochs. The curve for diffmpc was 2x shorter along the x-axis, so it showed a 2x speedup. For clarity, we have modified the figure to show losses as a function of epochs, illustrating that our method also converges on imitation learning problems. We now state the 2x faster training time in the text.
> 3. **Line search for RL experiment (Question 3)**: The reviewer is correct: in the RL experiment, the line search isn’t needed since the problem is convex. While we could have used the line search, we opted to disable it for all applicable solvers (DiffMPC, Trajax, mpc.pytorch) to isolate the algorithmic differences between methods, since the linesearch isn’t a new algorithmic contribution. The line search is enabled in the imitation learning and drifting experiments.
> 4. **Only inequality constraints are supported (Weakness 1)**: Please refer to our reply to Reviewer weUS regarding inequality constraints support in diffmpc: We discuss this reasonable limitation in the paper, cite relevant work to extend diffmpc in this direction, and now provide an initial implementation to directly handle inequality constraints via an augmented Lagrangian approach that shows speedups on the GPU.
> Only supporting equality constraints does not impact comparability to baselines:
> Like DiffMPC, the solver used in Trajax only natively supports dynamics-constrained problems.
> Mpc.pytorch supports an explicitly unconstrained problem, however, solving such a problem takes a code path with an inefficient list comprehension. Our minor modification speeds up mpc pytorch and ensures that we are testing algorithmic differences for solvers with equal capabilities, rather than implementation details.
> Theseus does not support equality or inequality constraints. Therefore, we roll the dynamics constraints into the cost, as is done in other peer-reviewed works using Theseus.
> 5. **Comparisons of Optimizers (Weakness 2)**: The reviewer is correct. We only compare against Trajax in the imitation learning and nonlinear attitude stabilization benchmarks because it is the fastest baseline in the convex RL experiment by 2x across a wide range of problems (Table 3).

---

> > ### Comment · Reviewer_4NTm · 2025-11-27
> >
> > Thank you for the clarifications which answer my questions. I will keep my positive rating.

---

### Official Review · Reviewer_cUhS · 2025-11-01

**Soundness:** 3
**Presentation:** 4
**Contribution:** 3
**Rating:** 8
**Confidence:** 3

**Summary:**

This paper introduces DiffMPC, a GPU-accelerated differentiable optimization solver for model predictive control (MPC) written in Jax. DiffMPC uses sequential quadratic programming with a line search and preconditioned conjugate gradient (PCG) with tridiagonal preconditioning to exploit the structure of their optimal control problem (OCP) for efficient GPU parallelization. Based on their formulation, DiffMPC benefits from warm-starting for both the forward and backward passes. Experimental results against existing solutions for GPU-accelerated differentiable MPC demonstrate that DiffMPC substantially benefits from GPU acceleration compared to baselines. DiffMPC is also demonstrated for real-world vehicle control.

**Strengths:**

- The paper is extremely well written and easy to follow. The authors provide comprehensive discussion on prior work in differentiable optimization and clearly explain their approach for differentiable MPC.

- DiffMPC is 2-7x faster than Trajax, a comparable Jax-based differentiable MPC library, and two PyTorch-based differentiable MPC libraries, Mpc.pytorch and and Theseus. Surprisingly, the authors also show in Figure 3 that Mpc.pytorch and Theseus obtain similar runtimes when running on both CPU and GPU, so their implementations do not benefit from GPU-acceleration, while DiffMPC does.

- The authors thoroughly explain the limitations of the proposed DiffMPC approach in Section 6. It is great to see this upfront, and the wider community may benefit when trying to build on the provided open source implementation. Based on a brief look at the Github repository, the code also appears to be well organized and easy to understand.

- DiffMPC is deployed for real-world control of a vehicle drifting through water puddles, to showcase practical use of the approach.

**Weaknesses:**

- DiffMPC does not directly handle boundary conditions.

- DiffMPC is slower on CPU than GPU compared to existing solutions, since DiffMPC avoids sequential Riccati recursions to better leverage GPU parallelism.

- There is no evaluation on the quality of the gradients computed by DiffMPC. Some analysis comparing between DiffMPC and finite difference gradients would be helpful.

**Questions:**

L404: Writing that DiffMPC is used within reinforcement learning is a little confusing. This makes it seem like this experiment is doing something similar to DiffTORI (Wan et al., 2024), which uses Theseus for differentiable trajectory optimization within a model-based RL approach with a learned world model. Based on my understanding, Section 5 is not doing this, but just directly using DiffMPC with a given differentiable model?

L484: How sensitive is DiffMPC to the initial guesses to PCG? How much tuning was done?

---

> ### Author Response · Authors · 2025-11-24
>
> Thank you for the constructive and thoughtful feedback. We hope that the following responses address your concerns.
> 1. **DiffMPC for RL (Question 1)**: The reviewer’s understanding is correct. We cite DiffTORI in our work, and show speedups of over 19x compared to Theseus (Fig. 3). We have clarified the language in Section 5 to clarify the setup of the RL experiment.
> 2. **Sensitivity to initial guesses for PCG (Question 2)**: We found that DiffMPC is not sensitive to the initial guess for PCG, and we always use a zero initial guess unless a warm-start is available. The effect of warm-starting PCG is evaluated in Appendix B, where we show marginal speedups from providing initial guesses to PCG from previous solves.
> 3. **Directly handling boundary constraints (Weakness 1)**: Please refer to our reply to Reviewer weUS: We explicitly discuss this limitation in the paper, cite relevant work to extend diffmpc in this direction, and provide an initial implementation to directly handle inequality constraints via an augmented Lagrangian approach that already shows significant speedups on the GPU compared to the CPU when increasing the batch size.
> 4. **CPU Performance (Weakness 2)**: The reviewer is correct: DiffMPC is designed for GPU use, and suboptimal performance on the CPU is an acknowledged limitation. However, we note that DiffMPC still has competitive performance on the CPU, on par with mpc pytorch and faster than Theseus. Also, performance on the CPU would improve with a C++ implementation, especially if using multithreading, which our JAX implementation doesn’t support on the CPU.
> 5. **Accuracy of gradients (Weakness 3)**: Please refer to our reply to Reviewer weUS: We have added a figure in Appendix B reporting cosine similarity between the computed gradients and gradients computed via finite differences for different PCG tolerances.

---

> > ### Comment · Reviewer_cUhS · 2025-11-26
> >
> > Thanks for the explanations. I have no further questions and will keep my score.

---

### Official Review · Reviewer_weUS · 2025-11-02

**Soundness:** 3
**Presentation:** 3
**Contribution:** 3
**Rating:** 6
**Confidence:** 3

**Summary:**

The paper introduces DiffMPC, a GPU-accelerated, differentiable model predictive control (MPC) solver built in JAX. The core idea is to solve and differentiate through optimal control problems (OCPs) using an SQP loop whose linear KKT systems are handled by a preconditioned conjugate gradient (PCG) routine with a symmetric stair (block-tridiagonal) preconditioner, exposing parallelism over time for both forward and backward passes. Warm-starting and reusing forward KKT factorizations in the backward VJP further reduce cost. Empirically, DiffMPC achieves 4–7× speedups over Trajax/mpc.pytorch/Theseus on GPU for RL/IL benchmarks, trains ~2× faster than Trajax on a nonlinear imitation-learning cart-pole, and supports an RL domain-randomization pipeline that improves robustness, with successful real-vehicle transfers.

**Strengths:**

1. The PCG with symmetric stair preconditioning cleanly leverages the OCP’s block-tridiagonal structure; both forward (solve) and backward (VJP) reuse the same KKT structure and benefit from warm-starts. The data-flow diagram (Fig. 2) and derivations (Eqs. 5–11) are clear.

2. Across RL timing benchmarks, DiffMPC on GPU is ~4× faster than the best baseline for the shown problem and 4–7× across others. For IL (cart-pole), training progress per wall-clock roughly doubles vs Trajax.

3. The RL+domain-randomization pipeline learns cost and tire parameters that eliminate spinouts in simulation (70% to 100% success) and transfer to a real Toyota Supra drifting donuts through puddles, illustrating robustness and policy generalization from differentiable MPC.

4. The paper documents solver choices including line search schedule, tolerances, and reports warm-start gains, explains baseline configurations.

**Weaknesses:**

1. RL timings disable line search and cap all solvers to a single iteration on convex QPs. mpc.pytorch is also modified to remove a serial list comp. While justified for timing, this setting may favor DiffMPC’s parallel linear algebra over sequential Riccati recursions. A complementary benchmark with full nonconvex solves would strengthen claims.

2. Warm-start gains are modest at strict tolerances (depending on $\epsilon$), and CPU performance is acknowledged to be worse. A more systematic analysis of the relation between batch size/horizon/tolerance and speed/accuracy/memory would clarify operational regimes.

3. Control bounds and other inequalities are handled outside OCP (e.g., in simulator) or via penalties, and the authors note future work for AL/IPM approaches. The absence of constrained OCP experiments limits conclusions for tasks where active sets matter.

**Questions:**

1. Can you report gradient-accuracy diagnostics, e.g., cosine similarity vs finite-difference or “exact” KKT-curvature gradients, on small OCPs, and show their effect on IL/RL convergence? This would directly test the stated trade-off in Sec. 3.4.

2. Could you include an experiment with explicit inequality constraints (e.g., torque/steering bounds) solved inside DiffMPC via an augmented-Lagrangian or interior-point variant, comparing robustness and speed to CPU solvers? Even a limited prototype would evidence extensibility.

3. For the drifting study, can you provide wall-clock training curves, GPU memory use, and ablations over batch size, horizon, and PCG tolerance, plus a sensitivity to differentiable vs non-differentiable simulator components? This would help others scope hardware needs and replicate behavior.

---

> ### Author Response · Authors · 2025-11-24
>
> Thank you for the constructive and thoughtful feedback. We hope that the following responses address your concerns.
> 1. **Accuracy of gradients (Question 1)**: As suggested, we have added a figure in Appendix B reporting cosine similarity between the computed gradients and gradients computed via finite differences. Results show the consistency of diffmpc’s computed gradients with finite differences across a range of PCG exit tolerances. While characterizing the convergence effects of learning with noisy gradients is beyond the scope of this work, for this problem, we found that exit tolerances smaller than or equal to 1e-6 lead to successful convergence.
> 2. **Extension to inequality constraints (Question 2 & Weakness 3)**: The solver can indeed be extended to handle inequality constraints via an augmented Lagrangian approach. We implemented an alternating direction method of multipliers (ADMM) scheme using diffmpc. Results on a nonlinear example with inequality box constraints in this notebook (github.com/diffmpc/diffmpc/blob/admm/admm.ipynb) show significant speedups on the GPU.  We will not include these results in the paper since they require additional work to rigorously study robustness in machine learning pipelines, and they go beyond our core contribution. As discussed in the conclusion, diffmpc can already handle inequality constraints via penalty functions (a standard approach in the literature, see for instance Equation 21 in https://arxiv.org/abs/2506.07823v2), and the control bounds could be accounted for in the dynamics via saturation functions. Our on-vehicle experiments indeed show strong robustness improvements using diffmpc, even though control inputs sent to the vehicle are bounded. In the conclusion, we also cite relevant works for future research.
> 3. **Operational Regimes (Question 3 & Weakness 2)**: We provide ablations over batch size and horizon (Table 3) and exit tolerances (Fig. 9 & Fig. 10) for the RL problems. For the drifting problem, we have added a section in Appendix B.3 with wall-clock training curves and reported memory use. Regarding memory use, we note that all results, including those for the drifting study, are collected on consumer-grade GPUs, emphasizing the practicality and accessibility of the method. We also note that memory utilization depends heavily on application-specific features like system dynamics and differentiation variables that determine the required space for the computation graph. Below, we include a table of GPU memory use when only differentiating with respect to the cost parameters (that are incorporated linearly into the optimization problem) or the dynamics parameters (that nonlinearly affect the problem definition).
>
> **GPU Memory Use (GB) Over Differentiation Variables**
> |  Batch Size  | Cost Parameters (nθ = 9) | Dynamics Parameters (nθ = 5) |
> |:---:|:--------------------:|:-------------------------:|
> |  8  |        0.6           |           4.7             |
> | 16  |        0.9           |           9               |
> | 32  |        2.5           |          17.7             |
>
>
> 4. **Benchmarking nonconvex problems (Weakness 1)**: The line search is enabled in the imitation learning and drifting experiments. For more information, please see our response to reviewer 4NTm. Regarding nonlinear examples, we do provide comparisons on a nonlinear imitation learning problem (Section 4.2) and a nonlinear attitude-stabilization problem (Table 4).

---

### Meta-Review · Area_Chair_tKSj · 2025-12-12

**Summary:**

This work proposes a new method that allows differentiable MPC to efficiently run on GPUs to exploit massive parallelization. This new approach demonstrates quite significant improvements over previous methods. The most remarkable part of the experiments is that the authors even conduct experiments in real car. I appreciate the authors' frankness by stating that the real car experiments are done only once due to hardware limitation so they refrain from drawing very strong conclusions from the real car experiments. All reviewers unanimously agree that the work makes a great contribution. I, therefore, recommend accept with oral presentation.

That being said, this paper is quite out of my field so my champion for an oral shouldn't be read as an expert opinion. If SAC and PC decide to accept this work as poster I am also ok with that.

**Reviewer Concerns:**

I think all concerns are addressed.

**Reviewer Scores:**

I think all reviewers will maintain their score.

---

### Decision · Program_Chairs · 2026-01-26

Accept (Oral)